# Exploring the molecular basis of the genetic correlation between body mass index and brain morphological traits

Daniela Fusco[1], Camilla Marinelli[1], Mathilde André[1], Lucia Troiani[1], Martina Noè[1], Fabrizio Pizzagalli[1], Davide Marnetto[1*], Paolo Provero[1,2*]

1 Department of Neurosciences "Rita Levi Montalcini", University of Turin, Turin, Italy, 2 Center for Omics Sciences, IRCCS Ospedale San Raffaele, Milan, Italy

* davide.marnetto@unito.it (DM); paolo.provero@unito.it (PP)

## Abstract

Several studies have demonstrated significant phenotypic and genetic correlations between body mass index (BMI) and brain morphological traits derived from structural magnetic resonance imaging (sMRI). We use the sMRI, BMI, and genetic data collected by the UK Biobank to systematically compute the genetic correlations between area, volume, and thickness measurements of hundreds of brain structures on one hand, and BMI on the other. In agreement with previous literature, we find many such measurements to have negative genetic correlation with BMI. We then dissect the molecular mechanisms underlying such correlations using brain eQTL data and summary-based Mendelian randomization, thus producing an atlas of genes whose genetically regulated expression in brain tissues is pleiotropic with brain morphology and BMI. Fine-mapping followed by colocalization analysis allows, in several cases, the identification of credible sets of variants likely to be causal for both the macroscopic phenotypes and for gene expression. In particular, epigenetic fine mapping identifies variant rs7187776 in the 5′ UTR of the *TUFM* gene as likely to be causal of increased BMI and decreased caudate volume, possibly through the creation, by the alternate allele, of an ETS binding site leading to increased chromatin accessibility, specifically in microglial cells.

## Author summary

Obesity is linked to many chronic diseases and its prevalence worldwide is increasing. Susceptibility to obesity is known to be due, to some degree, to genetic factors, and such genetic contribution is mediated in part by the brain through the control of food intake. Here we investigate the genetic variants that affect both obesity and brain morphology, identifying 21 genes whose expression affects both body mass index and morphological measures obtained from brain magnetic resonance imaging data. These results provide new insight into the complex relationships between the brain and obesity, also in view of

**Data availability statement:** GWAS summary statistics of brain imaging-derived phenotypes (IDPs) were obtained from the Oxford Brain

Imaging Genetics Server – BIG 40 – and specifically from https://open.win.ox.ac.uk/ukbiobank/big40/release2/stats33k/0001.txt.gz. GWAS summary statistics for BMI were obtained from the Neale Lab website (http://www.nealelab.is/uk-biobank), and specifically from https://broad-ukb-sumstats-us-east-1.s3.amazonaws.com/round2/additive-tsvs/21001_raw.gwas.imputed_v3.both_sexes.tsv.bgz Individual-level data were obtained from the UK BioBank (https://www.ukbiobank.ac.uk/) under application 86275. Individual-level data for GTEx subjects were obtained from dbGAP (https://www.ncbi.nlm.nih.gov/gap/) under project #18670. Genetic correlation data, gene/trait associations, and chromatin accessibility predictions produced for this work are available in the Supporting information.

**Funding:** This work was supported in part by a Fondazione CRT grant (grant 2021.1787 to P.P.). The funders had no role in study design, data collection and analysis, decision to publish, or preparation of the manuscript.

**Competing interests:** The authors have declared that no competing interests exist.

the recent development of revolutionary anti-obesity drugs, whose mechanisms of action are believed to include effects on the reward-related areas of the brain.

## Introduction

In 2022 one out of eight people were diagnosed with obesity worldwide, amounting to over 890 million adults and 160 million children and adolescents living with this chronic complex disease. In the majority of cases, obesity is a combination of environmental, psychosocial, and genetic factors. If the current increasing trend continues, 60% of the entire human population is estimated to be overweight or obese by 2030 [1].

The central nervous system has a role in susceptibility to obesity through the control of food intake [2]. This notion is strengthened by the fact that the heritability of body mass index (BMI), a parameter typically used by the World Health Organization to distinguish normal-weight ($18.5 \leq BMI \leq 25$) from obese people ($BMI \geq 30$), was found to be enriched in genes expressed in the brain and central nervous system [3,4]. In particular, a recent enrichment analysis of the genes located near 97 BMI-associated single-nucleotide polymorphisms (SNPs), aimed at identifying tissues and cell types with high expression of such genes, revealed that 27 out of 31 significantly enriched tissues were part of the central nervous system. However, these results are not enough to identify the specific brain regions involved [4].

Beyond the aforementioned results, several neuroimaging studies investigated the correlation between obesity and structural alterations in brain regions, especially differences in gray matter volumes. While the results of these studies have been somewhat conflicting, with both positive and negative correlations reported between gray matter volumes and obesity, most neuroimaging studies reported reduced gray matter volume in many brain regions, including several involved in executive control, to be associated with higher BMI [5]. In particular, reduced volume of cerebellum [6,7], basal ganglia [8], putamen [7], prefrontal cortex, temporal lobes, and subcortical structures [9], as well as reduced cortical thickness, were found to be associated with higher BMI [10–13].

On the other hand, a positive correlation between gray matter volume and BMI was found in nucleus accumbens and hypothalamus [14], while a twin study identified positive correlation with ventromedial prefrontal cortex and the right cerebellum [9].

The relationship between BMI and white matter integrity is more difficult to characterize since alterations show a more complex pattern [15]. Studies involving diffusion tensor imaging (DTI) investigated the influence of obesity on white matter integrity by comparing BMI with DTI parameters in adults, revealing a negative correlation between several microstructure architectures of the white matter, such as in the corpus callosum, and BMI [15].

To date, most studies have focused on phenotypic correlation. Regarding genetic correlation, a study explored the effects of a collection of obesity-related SNPs on 164 regional brain volume traits from the UK Biobank, finding that 17 such SNPs were associated with 51 regional brain volumes (both positively and negatively) [16]. A bivariate linkage and quantitative analysis on Mexican American individuals looked for genetic factors associated with increased BMI and reduced cortical surface area and subcortical volume, localizing two genome-wide significant QTLs at 17p13.1 and 3q22.1. The former pleiotropically affects ventral diencephalon volume and BMI, suggesting the involvement of such region in obesity through leptin-induced signaling in the hypothalamus. The latter involves the surface area of the supramarginal gyrus and BMI and might be relevant to the food-related reward mechanism [17].

Finally, a study integrating single-cell-RNA-sequencing and genome-wide association studies (GWAS) has shown that susceptibility to obesity is enriched for some hypothalamic cell types such as VMH Sf1-expressing neurons, most of which are involved in the integration of sensory stimuli, learning and memory. Despite neuroanatomical differences, these brain cell types share transcriptional patterns related to obesity [18].

Here, we systematically investigated the genetic correlation between BMI and brain morphology using GWAS summary data from the UK Biobank. To investigate the molecular mechanisms behind these correlations, we used summary-based Mendelian randomization (SMR) to identify the genes whose genetically regulated expression (GReX) in brain tissues is pleiotropic with BMI and brain morphology, suggesting a mediating effect. Finally, we used fine-mapping and colocalization to identify cases in which a single variant is causal of gene expression, BMI, and brain morphology, and epigenetic fine-mapping based on single-cell chromatin accessibility data to formulate mechanistic hypotheses on the relevant regulatory mechanisms and cell types.

## Results

### Genetic correlation

In order to assess the shared genetic basis between brain morphology and BMI, we relied on variant-trait associations for 435 cortical and subcortical measurements estimated on more than 30 thousand donors from the UK Biobank [19]. These measurements included volume, thickness, areas from white and pial surfaces, gray-white matter contrast for 54 cortical regions [20]; volumes and mean intensities for subcortical segments [21] (see Methods for details on the selection of traits).

Adopting cross-trait LD Score regression [22], the state-of-the-art summary-based approach, we found 108 nominally significant ($P < 0.05$) genetic correlations ($r_g$) across all categories. Most of the significant correlations turned out to be negative (Fig 1A), including in particular those between BMI and cortical areas (white and pial surfaces), cortical and subcortical volumes (all genetic correlation values are reported in S1 Data). The most significant correlations ($P < 0.001$ in at least one hemisphere) were all negative (Fig 1B), suggesting that, overall, genetic variants associated with increased BMI are also related to reduced size of brain structures. This is evident from the global subcortical volume and the globus pallidus in particular, and the total areas of pial and white surfaces, with especially negative $r_g$ values in cortical regions adjacent to the central sulcus, and in the temporal and cingulate lobes (Fig 1B). As expected, results for the two hemispheres are largely consistent with each other (Fig 1C).

Subcortical intensity traits, as provided by the UK Biobank (UKB), were analyzed, revealing a significant negative genetic correlation for brainstem intensity ($P<0.001$). However, interpreting this association is not straightforward. Although intensity values partially reflect what voxel-based morphometry (VBM) measures, VBM leverages the optimized intensity contrast among gray matter (GM), white matter (WM), and cerebrospinal fluid (CSF) tissues from T1-weighted images to classify voxels and estimate tissue volumes. Voxel intensity itself, however, can be influenced by MRI scanner artifacts and other sources of noise, introducing biases which are difficult to control. To address this, studies employing this method typically perform statistical analyses after normalizing tissue intensities, thereby estimating the contrasts between different tissues [23].

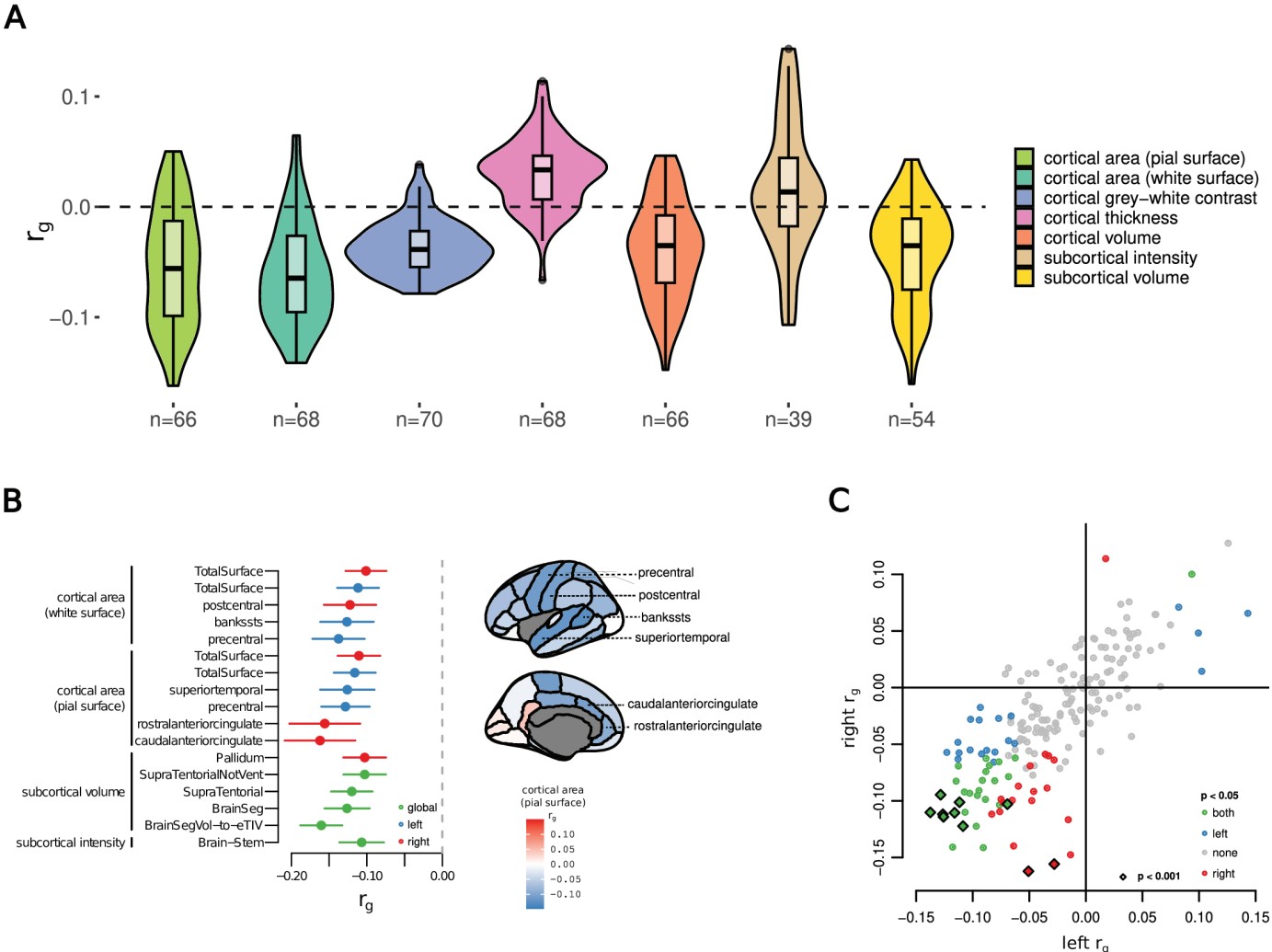

**Fig 1. Genetic correlations between sMRI traits and BMI.** (A) Genetic correlations ($y$ axis) of all 435 sMRI traits with BMI separated by category. $n$, number of traits for each category. (B) Traits with the most significant ($P<0.001$) genetic correlation ($x$ axis) with BMI, grouped by category. A few subcortical volume traits were removed to avoid redundancy. Measurements refer to the left (blue) or right (red) emisphere, or are global (green). Significant cortical regions are indicated on the brain surfaces on the right, which are colored according to their genetic correlation between pial surface area and BMI (left-right averages). (C) Comparison between genetic correlations in left ($x$ axis) and right ($y$ axis) hemispheres. Dots are colored based on their significance level. Diamonds represent correlations with $P<0.001$ in either or both hemispheres.

https://doi.org/10.1371/journal.pntd.1011658.g001

## Summary-based Mendelian randomization

We hypothesized the existence of molecular mechanisms at the level of gene expression underlying these genetic correlations. Mendelian randomization [24] can help elucidate the causal effect of an exposure on an outcome. Thanks to large databases of expression quantitative trait loci (eQTLs), gene expression can be used as an exposure, allowing the identification of putative genes whose regulation is pleiotropically related to a complex trait, suggesting a mediation effect of their expression. We used summary-based Mendelian randomization (SMR) [25] with gene expression in each of 13 brain tissues included in the GTEx database [26] as the exposure and sMRI traits and BMI as outcomes, reasoning that genes

associated with both sMRI and BMI in such analysis could provide a basis for the investigation of the molecular mechanisms driving the genetic correlation between the two types of traits.

SMR is unable, by itself, to distinguish between a causal relationship and a pleiotropic one, that is when gene expression and complex trait are influenced by the same genetic variants but not causally related. In addition, it can identify relationships which are actually mediated by different causal variants for exposure and outcome, when they are in linkage disequilibrium (LD) with each other. For this reason, SMR is often coupled with a test for heterogeneity in dependent instruments (HEIDI) [25], which can rule out a LD-mediated relationship.

A SMR + HEIDI analysis using the 435 sMRI traits and 13 GTEx brain tissues [26] found several genes potentially mediating the shared genetic components of BMI and sMRI traits through their expression (Fig 2A, respectively blue and green dots). A total of 21 genes (Fig 2A, red dots; S2 Data) were associated with both BMI and one or more sMRI traits.

We confirmed the gene/trait associations found by SMR through an individual-level study of the association between GReX and traits, as in transcriptome-wide association studies (TWAS) [27]: the genetic component of gene expression for the 21 genes was predicted in the relevant brain tissues for the UKB subjects for which the relevant phenotype was available (BMI or sMRI), and its correlation with the trait was computed. For BMI the TWAS association was nominally significant in 23 out of 34 cases, and the sign of the correlation was concordant with the one found by SMR for all of them. For sMRI traits, nominal significance and sign concordance was achieved in all of the 687 tested associations.

We asked whether the overlap between BMI- and sMRI-associated genes was greater than expected by chance. Since the significant genes for all traits are clearly clustered into genomic loci (see e.g. the loci in chromosomes 8 and 16 in Fig 2A, the latter including the well-known obesity associated *FTO* gene [28]), we devised a circular permutation procedure preserving the gene order (see Methods) which revealed that the overlap between BMI- and sMRI-associated genes is indeed greater than expected by chance (empirical $P < 10^{-4}$).

Furthermore, the cis- regions around the pleiotropic genes were enriched in genetic covariance between BMI and sMRI traits, although only nominally and for just 10 sMRI traits out of 97 tested (Monte-Carlo $P$-value from partial genetic covariance estimates). Nevertheless, as shown in Fig 2B, when considering the genetic regions surrounding all genes mediating BMI, i.e. without restricting to the 21 found to also mediate sMRI traits, we could not observe any significant enrichment: their estimated log-fold changes were markedly smaller and never distinguishable from 0. This suggests that the portion of the genome surrounding the 21 identified pleiotropic genes, albeit small, indeed carries a larger-than-expected proportion of the shared genetic determinants of brain morphology and BMI.

As shown in S1 Fig, there is a dense network of genetic correlations across the measurements from different brain regions for which we found pleiotropic genes with BMI. This is in line with a common genetic basis across many sMRI traits: it is therefore non-trivial to quantitatively pinpoint which morphological traits are the most affected by these genes. However, all categories show at least one pleiotropic gene (Fig 2C), and even though cortical gray-white contrast measurements seem very prominent, this can be readily explained by an especially strong genetic correlation among them (S1 Fig). Similarly, all GTEx tissues are involved, although with a suggestive prominence of subcortical tissues.

SMR also provides an estimate of the direction and size of the effect of gene expression on each trait. We observed that in most cases the effects of gene expression on BMI and sMRI traits have opposite signs (S2 Fig), in agreement with the prevalence of negative genetic correlations. Taken together, these results suggest that the genetic correlations between BMI and morphological brain measures are indeed driven in part by regulatory variants acting

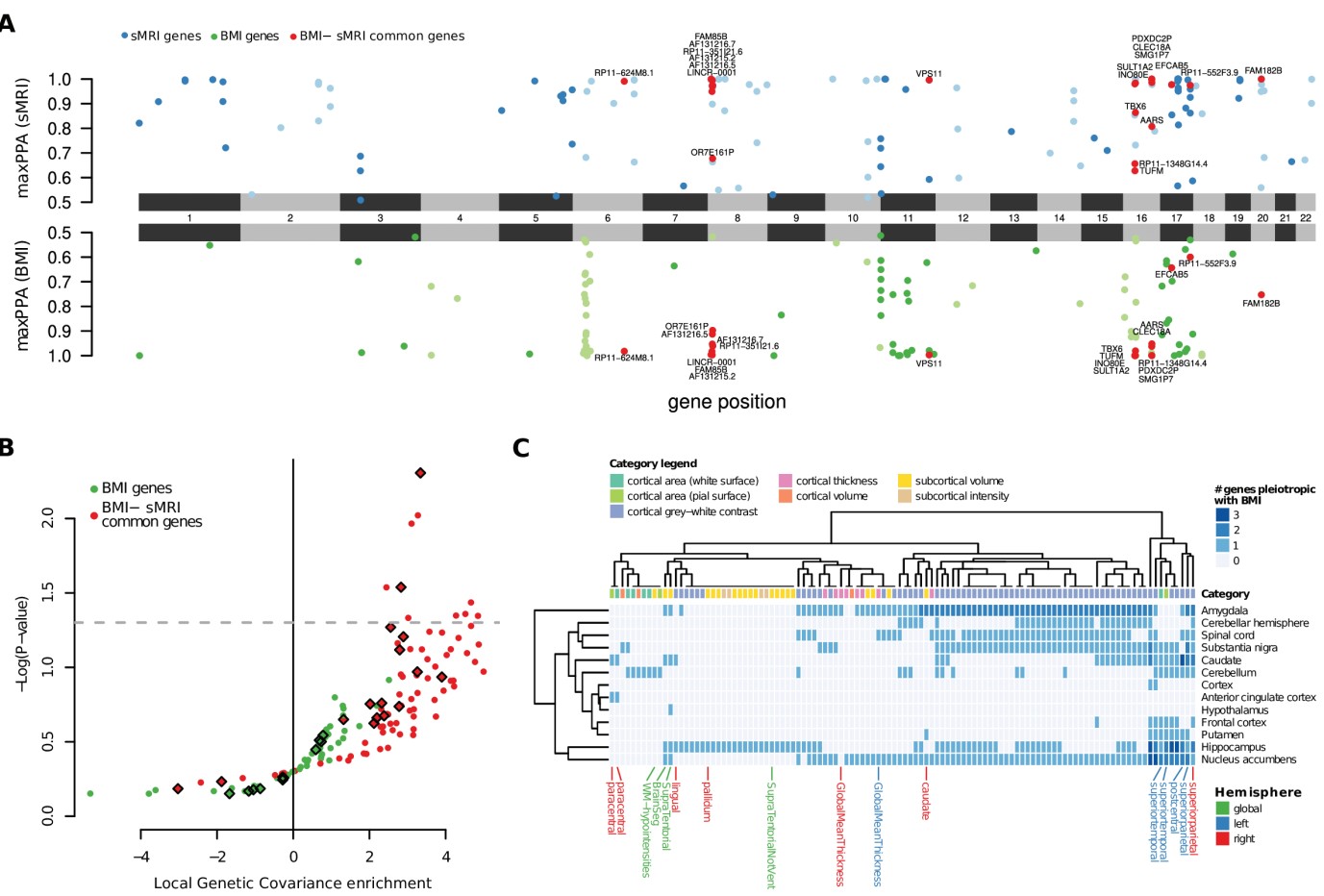

**Fig 2. SMR reveals common molecular patterns.** (A) Chicago plot showing the posterior probability of association (PPA, *y* axis) of gene expression on sMRI traits (top) and BMI (bottom) according to SMR. The *x* axis represents the genomic position. For each gene-sMRI trait, the reported PPA is the highest among all tissue/trait combinations. Red dots represent genes whose expression is associated with BMI and at least one sMRI trait in at least one tissue. (B) Volcano plot showing the local genetic covariance enrichment of each sMRI trait and its respective p-value. The enrichment is referred to the genomic regions surrounding the genes found by SMR to be in pleiotropy with BMI (green dots) or with both BMI and sMRI traits (red dots). Diamonds represent sMRI traits with significant (*P*<0.001) genetic correlation with BMI. (C) Heatmap representation of the number of genes whose genetically determined tissue expression pleiotropically affects BMI and each sMRI trait. sMRI traits refer to the left (blue) or right (red) emisphere, or are global (green)

https://doi.org/10.1371/journal.pntd.1011658.g002

on brain gene expression, and provide us with a list of candidate genes for the more detailed variant-level investigations described in the following.

## Fine-mapping and colocalization analysis

SMR reveals genes associated to both brain morphology and BMI through their expression in brain tissues, but does not provide direct information about the genetic variants involved. Although the heterogeneity test should rule out non-colocalizing associations between gene expression variation and complex traits (but not necessarily between BMI and sMRI), we used colocalization analysis to determine in which cases the variants likely to be causal of gene expression variation, BMI, and brain morphology coincided. Colocalization was investigated for all the BMI/sMRI trait/gene expression trios found through SMR analysis. Specifically, given such a trio, we used the Sum of Single Effects (SuSiE) regression framework combined with the COLOC algorithm (coloc-SuSie) [29], which avoids the assumption of

a single causal variant for each trait in each locus, to test colocalization for the three pairs of traits (gene expression - sMRI, gene expression - BMI, and BMI - sMRI). The results are shown in Fig 3, as a network whose nodes are traits, and edges indicate colocalized causal variants.

Of particular interest are cliques involving BMI, an sMRI trait, and the expression of a gene. In these cases a single variant is likely to be causal for the three traits. In particular, this happens for the protein-coding genes *VPS11* and *TUFM*, and the antisense transcripts *RP11-624M8.1* and *RP11-552F3.9*. For other genes, colocalization is limited to gene expression and sMRI traits. Note that the HEIDI test we used after SMR is also a test of colocalization between gene expression on one hand and BMI or sMRI traits on the other, which was passed by all the trait pairs shown in the figure. The fact that not all of them pass the coloc-SuSiE test indicates that the results of HEIDI and coloc-SuSiE agree only partially, a fact previously noted in [30]. Thus we conservatively consider the cliques in the network as having strong evidence of three-way colocalization, and the other phenotypic trios as having weak evidence of colocalization.

The Manhattan plots (Fig 4) show the SuSie 95% credible sets for the three traits for the TUFM and VPS11 loci (the loci associated to the two antisense transcripts are shown in S3 Fig). In both cases, as expected, the three-way intersection of the SuSie credible sets contains variants in high LD with each other ($R^2 > 0.5$). Notably, such intersections overlap the TUFM and VPS11 gene loci. Coloc analysis of each pair of traits showed strong evidence of colocalization (posterior probability of colocalization (PPH4) > 0.8).

## Epigenetic fine-mapping

While colocalization analysis strongly suggests that the same variant is causal of gene expression, BMI, and brain morphology for the four loci described above, it cannot identify the individual causal variant: Indeed (Fig 4, S3 Fig), in each of these loci, the intersection of the three credible sets still contains a sizable number of variants.

In order to further restrict the set of candidate causal variants for each locus, we investigated the effect of each of the variants included in the intersection of the credible sets of the three traits (gene expression, BMI, and sMRI) on chromatin accessibility in brain cells. This analysis was limited to the four loci with strong evidence of three-way colocalization, for which the intersections of the credible sets included a total of 128 variants, henceforth referred to as "candidate pleiotropic variants".

To this purpose, we first trained a gapped k-mer support vector machine (gkm-SVM [31]) on single-nuclei ATAC-seq data of 107 brain cell types produced in [32]. For 105 out of 107 cell types we had sufficient data to train a SVM, which we then used to predict the cell type-specific effect of each candidate pleiotropic variant on chromatin accessibility. Nine of the 128 candidate pleiotropic variants were predicted (|deltaSVM| > 2) to affect chromatin state in at least one brain cell type (S3 Data).

The strongest epigenetic fine mapping prediction concerned variant rs7187776, located in the 5' UTR of the *TUFM* gene (Fig 5A), and previously associated by GWAS to various complex traits including hip circumference adjusted for BMI [33]. The alternate allele was predicted by gkm-SVM to be associated with increased chromatin accessibility specifically in microglial cells (deltaSVM = 3.17). Indeed the same variant was found to be positively associated ($P = 1.1 \cdot 10^{-4}$) with chromatin accessibility in glia, but not neurons, in a recent study of the genetic determinants of chromatin accessibility in the human brain [34].

We thus decided to investigate in greater depth the possible mechanisms by which the alternate allele of rs7187776 could promote the opening of the chromatin in microglia.

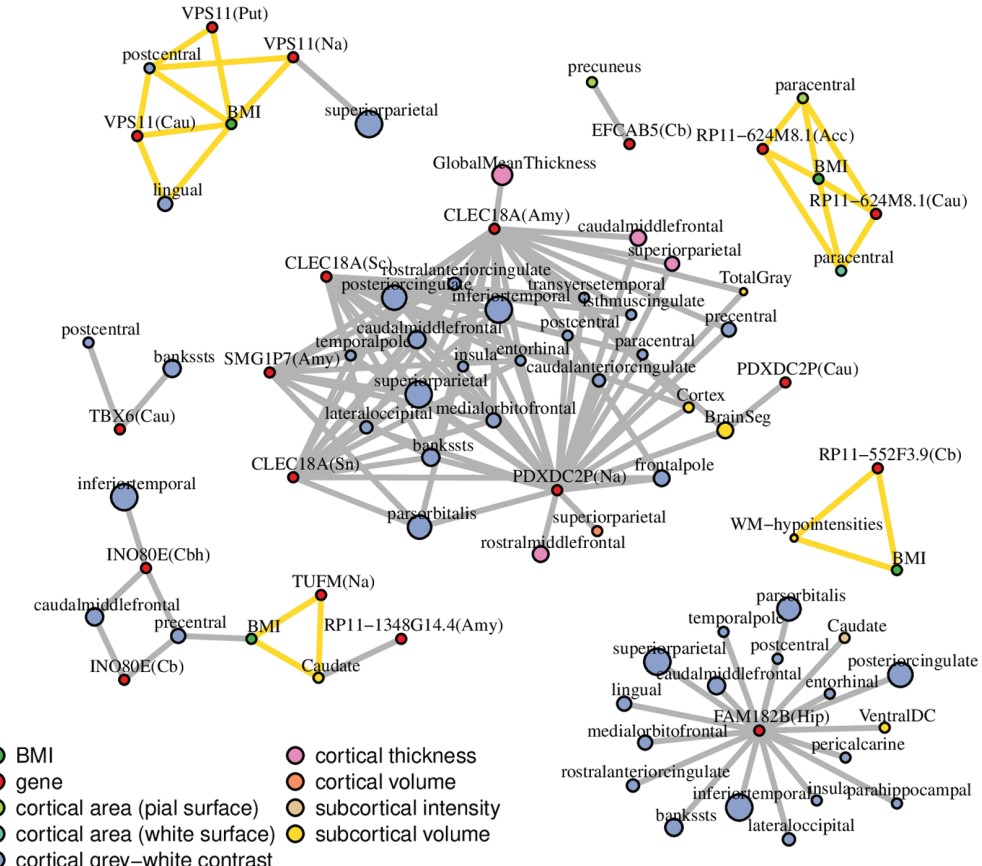

**Fig 3. Network of colocalized causal variants.** Each edge represents an instance of colocalization, that is a pair of causal variants (or rather genetic components tagged by such variants) for which the hypothesis of a common causal variant (H4) is the most likely according to COLOC; edges highlighted in yellow form a colocalization clique between eQTL, BMI and sMRI causal variants. Each node represents a trait: red nodes stand for gene expression in a GTEx tissue, green nodes for BMI, while sMRI nodes are colored according to their trait category. Traits can be duplicated if multiple independent causal variants are found for them. sMRI traits belonging to the same category with genetic correlation higher than 0.9 were pruned to leave only a representative node, sized proportionally to the trait count that it represents. For all left-right hemisphere pairs this pruning left a single representative node.

https://doi.org/10.1371/journal.pntd.1011658.g003

We used motifbreakR [35] to identify transcription factors (TFs) whose motif was altered by rs7187776. The analysis revealed that the alternate allele introduces a strong binding site for TFs of the ETS family, by creating the core ETS motif "GGAA" in lieu of "AGAA" found in the reference sequence. Specifically, motifbreakR identified several TFs whose affinity is predicted to be increased by the alternate allele, all belonging to the ETS family (Fig 5B,C).

Thus we hypothesized that rs7187776 opens the chromatin near the *TUFM* transcription start site (TSS) by creating a binding site for an ETS TF. To gain insight into the precise identity of such TF we used single-cell RNA-seq data curated by the Human Protein Atlas [36] to identify which of these TFs are expressed in microglia (Fig 5D). ETV6 and FLI1 showed the most robust expression in microglial cells, with a remarkable level of cell type-specificity especially for the latter.

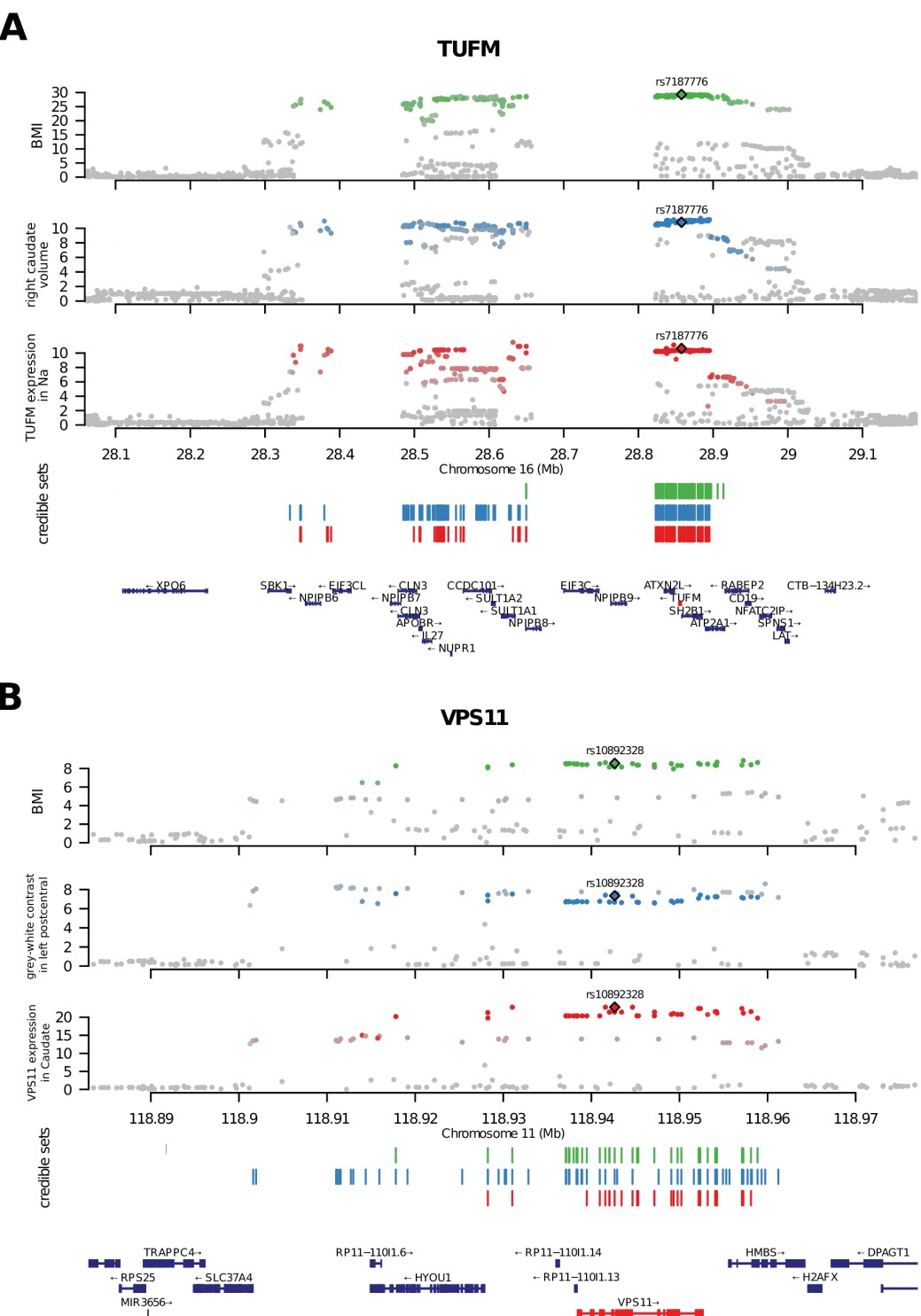

**Fig 4. Credible sets and evidence of colocalization.** Manhattan plots showing colocalization in the TUFM (A) and VPS11 (B) loci among the trait trios. Variants are plotted with their respective GWAS $-log_{10}(P)$ (see *y* axis). BMI (green), sMRI (blue), and gene expression (red) credible sets are shown by vertical bars. For each gene, the three-way intersection of the SuSie credible sets is tagged by a genome-wide significant variant represented by a diamond. Other variants in each credible set are represented by dots whose color represents the level of LD with the tagging variant (gray: $R^2 < 0.5$; colored: $R^2 > 0.5$, with color intensity proportional to $R^2$). Na= Nucleus accumbens.

https://doi.org/10.1371/journal.pntd.1011658.g004

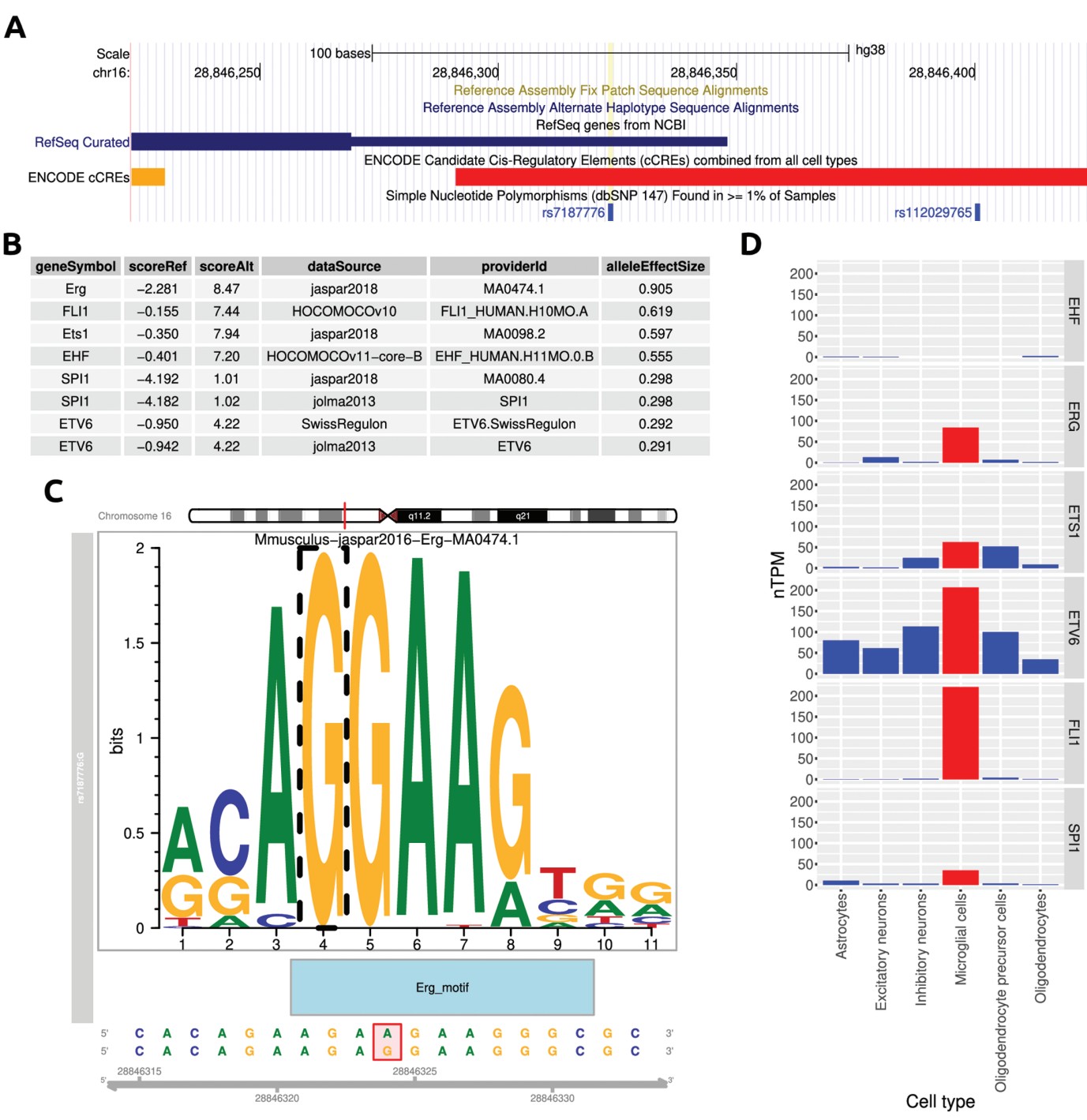

**Fig 5. Epigenetic fine mapping and motif analysis for the TUFM locus.** (A) Variant rs7187776 in the 5' UTR of TUFM is predicted by gkm-SVM to affect chromatin state specifically in microglial cells, with the alternate allele G associated with higher accessibility. (B) Motif analysis identifies eight motifs, corresponding to six transcription factors belonging to the ETS family, whose binding is strongly favored by the alternate allele. (C) The alternate allele creates the core ETS motif GGAA, here shown in the context of the ERG binding site. (D) The expression levels of the six transcription factors in brain cell types from the Human Protein Atlas: several of them are expressed in microglial cells, FLI1 and ETV6 showing especially robust and cell type-specific expression.

https://doi.org/10.1371/journal.pntd.1011658.g005

## Discussion

We have devised a strategy to dissect the molecular underpinnings of the genetic correlation between a macroscopic complex trait (BMI) and a set of endophenotypes (brain morphology parameters assessed by sMRI). Since the effects of genetic variants on complex traits is thought to be mediated mostly by gene regulation, we first used SMR to generate a catalog of 21 genes whose genetically regulated expression pleiotropically affects both the endophe-notypes and BMI. The enrichment of both the number of genes found and of the contribution of the respective genomic loci to the genetic correlation suggests that this approach can indeed explain at the molecular level a sizable portion of the observed correlations. In four of the identified loci, colocalization analysis confirmed the three-way pleiotropic effect of the fine-mapped genetic variants on gene expression, BMI, and brain morphology. For these loci we used epigenetic fine mapping to identify putative causal variants and the related mechanisms, highlighting in particular variant rs7187776, located in the 5' UTR of the *TUFM* gene and predicted to alter chromatin accessibility in microglia, possibly by creating a strong ETS binding site.

SMR identified genetically regulated *TUFM* expression to be positively associated with BMI and negatively associated with the volume of caudate in the right hemisphere. *TUFM* encodes a ubiquitously expressed nuclear-encoded mitochondrial protein, involved in mito-chondrial translation, that has been associated to several biological processes, including autophagy, and human phenotypes, including obesity (see [37] for a recent review). A study on subcortical volumes across lifespan has shown that *TUFM* expression in several brain regions in older and young adults is associated with caudate nucleus volume. In particular, increased *TUFM* expression was associated with smaller caudate nucleus volume, with evi-dence for colocalization in several tissues [38]. TUFM was also found associated with caudate volume and other subcortical morphological features in a recent GWAS meta-analysis [39].

Epigenetic fine mapping and motif analysis suggest a mechanism by which variant rs7187776 in the 5' UTR of *TUFM* creates a strong ETS binding site resulting in microglia-specific chromatin opening. Of note, chronic microglia activation has been associated with decreased hippocampus and parahippocampus volume in neurodegenerative diseases [40]. According to dbSNP, the variant is common in all populations ascertained, with minor allele frequency (MAF) between 0.11 and 0.48. Since the magnitude and direction of causal eQTL effects are known to be consistent across populations [41], it is reasonable to expect that the findings related to this SNP should be transferable to other populations, although we did not formally check whether this is the case.

Other three-way colocalization signals involved BMI, *VPS11* expression in various sub-cortical structures, and gray-white contrast phenotypes from sMRI. *VPS11*, a subunit of the CORVET complex involved in the endosome/lysosome pathway, has an essential role in brain white matter development and neuron survival in zebrafish, while its reduced expres-sion leads to an impaired autophagic activity in human cells [42]. To the best of our knowl-edge it has not been investigated in the context of BMI and obesity, although GWASs found *VPS11*-associated variants associated to BMI [43] and HDL cholesterol measurements [44].

The remaining high-confidence colocalization signals involved two non-coding RNAs, namely *RP11-552F3.9* and *RP11-624M8.1* (also known as *HEY2-AS1*), antisense transcripts of *TRIM47* and *HEY2*, respectively. Variants mapped to these genes have been associated by GWASs to both brain morphology traits [19,45] and BMI [46,47]. Indeed *TRIM47* was among the genes found in [16] to harbor variants associated with both BMI and regional brain volumes.

## Limitations

Some limitations of this work should be noted. It is possible that some of the pleiotropy signals were lost by summarizing the genetic basis of our complex traits of interest into effect-sizes and applying summary-based analyses. While the high number of sMRI traits analyzed initially warranted this approach, individual-based techniques capable of dealing with large sample sizes and multiple traits simultaneously are now being developed, opening the possibility of an individual-based analysis. Notably, the fact that the estimated GReX was predictive of BMI and sMRI traits for the vast majority of the genes identified by SMR, is instead reassuring against the risk of false positives.

Another limitation is that even if we set out to evaluate genetic correlation and pleiotropy for fine-grained brain morphology measurements, the strong genetic correlation across all the sMRI traits analyzed and the lack of an explicitly differential strategy prevented us from pinpointing specific brain regions consistently involved throughout all steps of our analysis.

Epigenetic fine-mapping was performed with the SVM-based methods of [31], while methods to predict open chromatin from sequence based on convolutional neural networks, such as Basset [48] or Sei [49], might in principle achieve higher predictive power. However, the method we used has the advantage of a much shorter training time, a key issue as it had to be applied to each of the more than 100 cell types identified in [32].

Finally, the use of BMI as an obesity-related quantitative trait should be complemented by other obesity-related measures, such as waist-to-hip ratio, body fat percentage, fat-free mass, etc. The analysis and comparison of the genetic correlations of these traits with sMRI measurements is likely to allow a more nuanced view of the relationship between obesity and brain morphology.

## Conclusion

We have shown how the integration of data from GWASs, population-based transcriptomic studies, and single-cell chromatin accessibility assays allows the molecular dissection of the genetic correlation between complex traits and the formulation of data-driven hypotheses on the relevant regulatory mechanisms. Specifically, we have confirmed and expanded previous observations on the genetic correlation between brain morphological features and BMI, and generated an atlas of 21 genes whose genetically regulated expression mediates a significant part of such correlation.

## Methods

### Summary statistics and individual data

We used the GWAS summary statistics of brain imaging-derived phenotypes (IDPs) processed with a sample size of about 33 thousands individuals (sample sizes vary across traits) from the UKB release 2020 [19]. In particular, we selected all the IDPs defined according to the Desikan-Killiany cortical atlas [20] and subcortical volumetric segmentation (ASEG) [21]. After excluding the traits "aseg_lh_number_HolesBeforeFixing" and "aseg_rh_number_HolesBeforeFixing," which correspond to technical artifacts produced by segmentation rather than morphological measurements, we were left with 435 IDPs (see S1 Data for a full list). GWAS summary statistics for BMI were derived from the UK Biobank GWAS version 3, and made available by the Neale Lab (http://www.nealelab.is/uk-biobank). All summary statistics for both IDPs and BMI were filtered for imputation INFO score ≥ 0.95 and minor allele frequency (MAF) ≥ 0.01.

For methods that required individual-level information (LD matrices, stratified LD scores, GReX), we used 38,406 samples of European ancestry from the UKB. Subjects were selected among the donors with brain MRI-derived phenotypes as of October 2023 (ASEG whole brain volume, ID 26514, used as representative trait for filtering) identified as British with West-European ancestry (field 22006, code 1), and not considered outliers for missingness and heterozygosity (field 22027). Finally, we pruned relatives with kinship coefficient ≥ 0.04419 (third or lower degree relatives), reaching a subset with the sample size mentioned above. A total of 9,252,961 variants with INFO score > 0.8 and minor allele frequency (MAF) > 0.01 were included in this dataset.

## Genetic correlation

We estimated the genetic correlation among each of the 435 sMRI-derived traits and BMI using GWAS summary statistics with the cross-trait LD Score regression (LDSC) software obtained from https://github.com/bulik/ldsc. We used genome-wide LD scores computed by the Pan-UK consortium on UKB samples of European descent (https://pan.ukbb.broadinstitute.org/): scores were available for 1,094,844 SNPs which met their exporting criteria, most notably after filtering for HapMap3 SNPs, MAF > 0.01 and removing the human major histocompatibility complex (MHC) region.

## Summary-based Mendelian randomization

We conducted summary-based Mendelian randomization with the Omics Pleiotropic Association (OPERA) software tool [50] to identify pleiotropic associations. Summary-level expression quantitative trait loci (eQTL) data of European individuals (n = 698) from the GTEx [26] dataset, including 13 brain tissues, were used for the analysis as the exposure. We estimated the cis-eQTLs with the R package Matrix eQTL [51], testing all variants with MAF ≥ 0.01 within 2 megabases (Mbs) of each gene's TSS. All the covariates included by GTEx for each tissue, including sex, genomic PCs, and PEER factors, were used in the regression models.

BMI and sMRI-derived traits were considered one by one as outcomes. We used 503 European samples from the 1000 Genomes Project [52] as the LD reference data required for the HEIDI test. We set the significance threshold of posterior probability of association (PPA) to 0.5, then performed the HEIDI test for the genes passing this threshold and discarded those for which the null hypothesis of non heterogeneity could be rejected with $P < 0.05$.

The circular permutation procedure to test for BMI/sMRI gene overlap was devised as follows: for 10,000 times, a list of all genes tested, ordered for TSS position and tagged for association status with BMI, was split into K genomic chunks and a random spin was applied to each chunk; chunks were then reassembled in a random order. This resulted in 10,000 random sets with the same positional clustering of the genes identified for BMI, which were used to compute an empirical $P$-value for the size of the intersection with genes associated with at least one sMRI trait. The procedure was repeated with K= 1,5,10, obtaining the same result.

## Individual-level GREx-trait association

An in-house implementation of prediXcan [27] was used to train an elastic net regression for the 21 putatively pleiotropic genes on the 13 GTEx brain tissues mentioned above. We used for training 80% of the European samples used for eQTL estimation, using 1 Mb as cis-distance from each gene's TSS and including all encompassed variants in the training. Gene expression was first transformed into residuals using the approach described in [30], then regressed on the genotype matrices. We filtered out models achieving Pearson's correlation

$r < 0.1$ between predicted and measured expression, or with zero variance, in the 20% hold-out testing set. Filtering on the signed correlation coefficient, rather than on its square, allows excluding negative correlations between predicted and actual expression, which clearly do not signal good predictive power [53]. The resulting models were applied on individual UKB genetic data described above, for which the GReX was imputed and used as predictor for the BMI and sMRI phenotypes identified by SMR+HEIDI analysis, together with sex and age as covariates.

### Partitioned genetic covariance

We aimed to test enrichment in genetic covariance in the genomic loci around two sets of genes: a) those found associated with both BMI and at least one sMRI trait by SMR analysis, and b) all genes associated with BMI by SMR. These regions were identified by considering a 4 Mb window surrounding the TSS of each gene in each set.

A cross-trait stratified LD Score Regression (sLDSC) [3] was conducted to estimate the partitioned genetic covariance between BMI and sMRI phenotypes at each of the two aforementioned annotations, using GWAS summary statistics as described previously. We limited this test to the 97 sMRI phenotypes previously found to be genetically correlated with BMI at nominal significance.

We used the individual-level UKB genetic data described above to compute partitioned LD scores, setting a 2 Mb LD window and keeping SNPs in HapMap3 and with MAF > 0.01 for the regression. Enrichments and the corresponding $P$-values were computed starting from the genetic covariance estimates and relative standard errors reported in the LDSC logs, by generating 50,000 Monte Carlo randomizations and testing how often the genetic covariance fraction was less or equal than the SNP fraction, thus obtaining one-sided $P$-values. The enrichment estimates were extracted from the medians of such distributions and then log2-transformed.

### Fine-mapping and colocalization

In order to assess three-way colocalization assuming multiple causal variants per locus, we applied coloc to the decomposed signals found by SuSiE (coloc-SuSiE [29]) for each locus whose expression in at least one brain tissue was found by SMR+HEIDI analysis to be in association with BMI and a sMRI phenotype. The analysis was performed on each pair of phenotypes resulting from the combination of BMI, sMRI traits and gene expression, then the colocalization results were compared across the three variables. Only SNPs with effect sizes available for all three phenotypes were considered. The *coloc_susie* function from the coloc 5.2.3 R package was run with default parameters. We computed the LD matrix for effects on gene expression with the GTEx European samples [26] from each tissue. The LD matrix for the effects on the complex trait from UKB was computed on the individual UKB genetic data described above. We considered that colocalization occurred when the colocalization posterior probability (PPH4) was greater than the probability of association with different causal variants (PPH3). Manhattan plots were produced with locuszoom [54].

### Epigenetic fine mapping and motif analysis

The open chromatin regions of the brain cell types identified in [32] were used to train gkm-SVM [31] models which were then used to predict the effect of each variant on the chromatin accessibility of each cell type. The analysis was performed on the 105 cell types (out of a total of 107 identified in [32]) for which at least 5,000 open chromatin regions were available.

Each model was trained on 5,000 open chromatin regions drawn at random from the complete set. This analysis was applied to the 128 variants included in the three-way intersection of credible sets (BMI, sMRI, and gene expression) for the four loci for which we had strong evidence of pleiotropy (supported by both HEIDI and COLOC).

Candidate transcription factor binding sites altered by the variants of interest were identified with motifbreakR [35], using the whole MotifDb collection of positional frequency matrices and log probability scores. We considered as significantly motif-altering the variants whose effect was evaluated as "strong" by motifbreakR and such that the logarithmic score was positive for one allele and negative for the other one.

## Supporting information

### Supporting Data

**S1 Data: Genetic correlation between BMI and 435 sMRI traits.** The columns contain:

- Pheno, UKB.ID : UKB phenotype codes
- ROI: region of interest
- Category.name: phenotype category
- rg: estimated genetic correlation
- se: standard error
- P-value: P-value of the null rg = 0

(XLSX)

**S2 Data: Genes associated by SMR to both BMI and at least one sMRI trait.** The columns contain:

- Gene_ID: GENCODE gene id
- Tissue: GTEX tissue
- Pheno: UKB code of sMRI trait
- p_HEIDI_sMRI: HEIDI P-value for the sMRI trait
- b_SMR_sMRI: effect size of GReX on sMRI
- PPA_sMRI: posterior probability of association between GReX and sMRI trait
- p_HEIDI_BMI: HEIDI P-value for BMI
- b_SMR_BMI: effect size of GReX on BMI
- PPA_BMI: posterior probability of association between GReX and BMI

(XLSX)

**S3 Data: Variants in the three-way intersections of credible sets predicetd to alter chromatin accessibility in brain cell types.** The columns contain:

- cell_type: as in [32]
- coordinates: SNP coordinates, hg38
- rsid: SNP id
- ref: reference allele
- alt: alternate allele
- delta_svm: deltaSVM value
- ENSG00000267801, ENSG00000178952, ENSG00000237742, ENSG00000160695: whether the SNP is included in the three-way intersection of credible sets for each gene

(XLSX)

## Supporting figures:

**S1 Fig. Genetic correlation between sMRI traits.** Only the correlations between traits in the same category were computed, plus those between cortical area (pial surface) and cortical area (white surface). ∗, nominally significant ($P < 0.05$).
(PDF)

**S2 Fig. Effect size of GReX on BMI and sMRI traits for the 21 common genes**.
(PDF)

**S3 Fig. Manhattan plots as in Fig 4 for the non-coding RNAs** *RP11-552F3.9* **and** *RP11-624M8.1*, **respectively antisense transcripts of** *TRIM47* **and** *HEY2*.
(PDF)

## Software used

- LDSC v1.0.1
- Matrix eQTL v2.3
- OPERA v1.0.0
- gkmSVM v0.80.0
- motifbreakR_2.18.0

## Acknowledgments

Daniela Fusco and Camilla Marinelli are PhD students enrolled in the National PhD in Artificial Intelligence (resp. XXXVIII and XL cycle) course on Health and Life Sciences, organized by Università Campus Bio-Medico di Roma. UKB data were accessed under application 86275. GTEx data were accessed from dbGAP under project #18670.

## Author contributions

**Conceptualization:** Fabrizio Pizzagalli, Davide Marnetto, Paolo Provero.

**Data curation:** Daniela Fusco.

**Formal analysis:** Daniela Fusco, Camilla Marinelli, Mathilde André, Lucia Troiani, Martina Noè, Davide Marnetto.

**Funding acquisition:** Paolo Provero.

**Investigation:** Daniela Fusco, Camilla Marinelli, Mathilde André, Lucia Troiani, Martina Noè, Davide Marnetto.

**Methodology:** Daniela Fusco, Camilla Marinelli, Mathilde André, Davide Marnetto, Paolo Provero.

**Software:** Daniela Fusco, Camilla Marinelli, Mathilde André, Lucia Troiani, Martina Noè, Davide Marnetto.

**Supervision:** Fabrizio Pizzagalli, Davide Marnetto, Paolo Provero.

**Visualization:** Daniela Fusco, Martina Noè, Davide Marnetto.

**Writing – original draft:** Daniela Fusco, Davide Marnetto, Paolo Provero.

**Writing – review & editing:** Daniela Fusco, Camilla Marinelli, Mathilde André, Lucia Troiani, Martina Noè, Fabrizio Pizzagalli, Davide Marnetto, Paolo Provero.

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
