## [Decision Letter · Decision Letter 0]

31 Jan 2025

PGENETICS-D-24-01224

Exploring the molecular basis of the genetic correlation between body mass index and brain morphological traits

PLOS Genetics

Dear Dr. Provero,

Thank you for submitting your manuscript to PLOS Genetics. After careful consideration, we feel that it has merit but does not fully meet PLOS Genetics's publication criteria as it currently stands. Therefore, we invite you to submit a revised version of the manuscript that carefully addresses the points raised during the review process.

Please submit your revised manuscript within 60 days Apr 01 2025 11:59PM. If you will need more time than this to complete your revisions, please reply to this message or contact the journal office at plosgenetics@plos.org. Please include the following items when submitting your revised manuscript:

We look forward to receiving your revised manuscript.

Kind regards,

Yun Li

Academic Editor

PLOS Genetics

Hua Tang

Section Editor

PLOS Genetics

Aimée Dudley

Editor-in-Chief

PLOS Genetics

Anne Goriely

Editor-in-Chief

PLOS Genetics

**Journal Requirements:**

At this stage, the following Authors/Authors require contributions: Daniela Fusco, Camilla Marinelli, Mathilde André, Lucia Troiani, Martina Noè, Fabrizio Pizzagalli, Davide Marnetto, and Paolo Provero. Please ensure that the full contributions of each author are acknowledged in the "Add/Edit/Remove Authors" section of our submission form.

The list of CRediT author contributions may be found here: https://journals.plos.org/plosgenetics/s/authorship#loc-author-contributions

https://journals.plos.org/plosgenetics/s/submission-guidelines#loc-parts-of-a-submission

5) When completing the data availability statement of the submission form, you indicated that you will make your data available on acceptance. We strongly recommend all authors decide on a data sharing plan before acceptance, as the process can be lengthy and hold up publication timelines. Please note that, though access restrictions are acceptable now, your entire data will need to be made freely accessible if your manuscript is accepted for publication. This policy applies to all data except where public deposition would breach compliance with the protocol approved by your research ethics board. If you are unable to adhere to our open data policy, please kindly revise your statement to explain your reasoning and we will seek the editor's input on an exemption. Please be assured that, once you have provided your new statement, the assessment of your exemption will not hold up the peer review process.

6) Please amend your detailed Financial Disclosure statement. This is published with the article. It must therefore be completed in full sentences and contain the exact wording you wish to be published. State what role the funders took in the study. If the funders had no role in your study, please state: "The funders had no role in study design, data collection and analysis, decision to publish, or preparation of the manuscript.".

**Reviewers' comments:**

Reviewer's Responses to Questions

**Comments to the Authors:**

Reviewer #1: Fusco et. al, in their manuscript “Exploring the molecular basis of the genetic correlation between body mass index and brain morphological traits” show their exploration of the molecular basis of genetic correlation between BMI and brain morphological traits integrating genomic, transcriptomic, and epigenetic data. The authors first found significant genetic correlations between BMI and brain morphology, primarily negative correlations, then identified 21 genes whose GReX in brain tissue is associated with both BMI and brain traits. TUFM and VPS11 are used as showcase examples. Further the authors did three-way colocalization and fine-mapping and reported four loci. Variant rs7187776 in the TUFM gene's 5′ UTR was highlighted as a potential causal variant influencing BMI and brain structure through changes in chromatin accessibility in microglial cells.

The analysis show novelty and is comprehensive, albeit the following concerns should be addressed:

(1) Why used different INFO score filtration for GWAS summary statistics for BMI and brain traits?

(2) Could you elaborate the reasons that you picked TUFM and VPS11 for showcase?

(3) Have you assessed how transferable are these findings to other population? What are the allele frequency of rs7187776 in European and other population?

(4) How did you select 435 sMRI traits? Are they purely based on data availability or selected by some criteria?

(5) Would including pathway analysis for TUPM and VPS11 may be helpful to uncover some biological process that may align with your finding?

(6) Most jargons and abbreviations are well-defined, while some were not at the first occurance. For example, “PPA” didn’t defined in Figure 2 legend and main text. In main text of coloc-SuSiE analysis, posterior probabilities (e.g. PPH3, PPH4), are introduced without introducing the hypothesis.

(7) Make sure the “x/y axis” in figure legend are consistent for italic or not. Typo for “axis” in Figure 4 legend.

(8) Specify version for softwares you used.

(9) “Method: Individual-level GREx-trait association”: R -> R^2

Reviewer #2: The study systematically examines the genetic correlations between body mass index and brain morphological traits using data from the UK Biobank. It identifies negative genetic correlations between BMI and multiple brain structure measurements. Through integrative approaches, including Mendelian randomization and fine-mapping, the research highlights genes, like TUFM, associated with these correlations. A notable finding is that specific genetic variants, such as rs7187776, potentially affect BMI and brain morphology by altering chromatin accessibility in microglial cells. I have such questions:

1. In Figure 1, could you please use subscripts for genetic correlations as is done in the main text?

2. For Figure 1b, what are the rg values for the points that exceed the left-side x-axis?

3. In Figure 2a, it would be helpful to include a legend to better differentiate it from Figure 2b. Additionally, do you have a threshold?

4 .How do you deal with the covariates like age and sex?

Reviewer #3: The manuscript used several analysis methods to explore the relationships among genetic features, BMI, and sMRI traits. The analyses first demonstrated negative genetic correlations between BMI and most sMRI traits. Then assuming gene expressions are the underlying causes of the genetic correlations. 21 genes were found associated with both traits. Colocalization and epigenetic fine mapping found a variant to be causal.

Results: summary based mendelian randomization

SMR + HEIDI and PrediXcan TWAS both identify genes associated with traits. Will PrediXcan identify different gene sets as the 21 found by SMR? Are the 21 genes found by SMR causal? Is this the reason why using SMR over PrediXcan?

Epigenetic fine mapping

Are there any other learning models can be applied to this context? If so, what is the advantage of gkm-SVM?

Colocalization

Explain more on why three way colocalization identified variants are causal?

**Have all data underlying the figures and results presented in the manuscript been provided?**

Reviewer #1: Yes

Reviewer #2: None

Reviewer #3: Yes

PLOS authors have the option to publish the peer review history of their article (what does this mean?). If published, this will include your full peer review and any attached files.

Reviewer #1: No

Reviewer #2: No

Reviewer #3: No

**Figure resubmission:**
---

## [Decision Letter · Decision Letter 1]

17 Mar 2025

Dear Dr Provero,

We are pleased to inform you that your manuscript entitled "Exploring the molecular basis of the genetic correlation between body mass index and brain morphological traits" has been editorially accepted for publication in PLOS Genetics. Congratulations!

Yours sincerely,

Yun Li

Academic Editor

PLOS Genetics

Hua Tang

Section Editor

PLOS Genetics

Aimée Dudley

Editor-in-Chief

PLOS Genetics

Anne Goriely

Editor-in-Chief

PLOS Genetics

Comments from the reviewers (if applicable):

Reviewer's Responses to Questions

**Comments to the Authors:**

Reviewer #1: The authors have well addressed my questions, and the novelty and comprehensive analysis are well aligned with the standards of PLoS Genetics.

Reviewer #2: The authors addressed my concerns.

Reviewer #3: My questions have been addressed.

**Have all data underlying the figures and results presented in the manuscript been provided?**

Reviewer #1: Yes

Reviewer #2: None

Reviewer #3: Yes

PLOS authors have the option to publish the peer review history of their article (what does this mean?). If published, this will include your full peer review and any attached files.

Reviewer #1: No

Reviewer #2: No

Reviewer #3: No

**Data Deposition**

http://datadryad.org/submit?journalID=pgenetics&manu=PGENETICS-D-24-01224R1

**Press Queries**

---

## [Editor Report · Acceptance letter]

PGENETICS-D-24-01224R1

Exploring the molecular basis of the genetic correlation between body mass index and brain morphological traits

Dear Dr Provero,

We are pleased to inform you that your manuscript entitled "Exploring the molecular basis of the genetic correlation between body mass index and brain morphological traits" has been formally accepted for publication in PLOS Genetics! Your manuscript is now with our production department and you will be notified of the publication date in due course.

With kind regards,

Anita Estes

PLOS Genetics

On behalf of:
